# IN DEFENSE OF SEMI-SUPERVISED LEARNING: SELF-SUPERVISION IS NOT ALL YOU NEED

## ABSTRACT

Self-supervised (SELF-SL) and Semi-supervised learning (SEMI-SL) are two dominant approaches in limited label representation learning. Recent advances in SELF-SL demonstrate its importance as a pretraining step to initialize the model with strong representations for virtually every supervised learning task. This "SELF-SL pretraining followed by supervised finetuning" pipeline challenges the benefits of SEMI-SL frameworks. This paper studies the advantages/disadvantages of SELF-SL and SEMI-SL frameworks under different conditions. At its core, this paper tries to answer the question "When to favor one over the other?". In particular, we explore how the choice of SELF-SL versus SEMI-SL framework affects performance in in-domain, near-domain and out-of-distribution data, robustness to image corruptions and adversarial attacks, cross-domain few-shot learning, and the ability to learn from imbalanced data. Surprisingly, contrary to popular belief, our extensive experiments demonstrate that in-domain performance and robustness to perturbations are the two biggest strengths of SEMI-SL approaches, where they outperform SELF-SL methods by huge margins, while also matching Self-supervised techniques on other evaluation settings.

## 1 INTRODUCTION

Large-scale annotated computer vision datasets such as ImageNet Deng et al. (2009) and Kinetics Kay et al. (2017) have been instrumental in remarkable progress towards solving many practical computer vision tasks such as object recognition He et al. (2016), object detection Ren et al. (2015), and image segmentation Chen et al. (2017). Nevertheless, due to the high cost of annotating large-scale datasets, label-efficient representation learning has been an active area of research in the vision community Chen et al. (2020a); Dave et al. (2022). Methods for learning label-efficient representations can be roughly grouped into two broad approaches: Self-Supervised Learning (SELF-SL) and Semi-Supervised learning (SEMI-SL). The SELF-SL approach aims to learn a generic task-agnostic visual representation using large unlabelled datasets, which can then be finetuned *later on smaller labelled datasets*. From a different point of view, SEMI-SL framework *introduces the available limited labels from the very beginning* of training, and rely on them to take advantage of a large unlabeled dataset. These two lines of research have progressed largely independent of each other. Nevertheless, they both aim to reduce annotation cost of large datasets through utilizing unlabelled data. In this work we study the differences between representations learned through comparable instantiations of these techniques to understand the advantages and disadvantages inherent in each choice. In particular, we employ notable SELF-SL methods such as SwAV Caron et al. (2020), and SimCLR Chen et al. (2020a), to compare against PAWS Assran et al. (2021) and SimMatch Zheng et al. (2022), which are representative SEMI-SL methods. We note that our study does not include every SELF-SL and SEMI-SL method, specifically, we do not include Masked autoencoder He et al. (2022) and DINO Caron et al. (2021b) because they are not directly comparable to current prominent SEMI-SL techniques.

We analyze different aspects of the representations learned by SELF-SL versus SEMI-SL to establish patterns that can aid practitioners in adopting the best framework to meet their requirements. The first criterion that we explore is *in-domain performance*, which is the classification accuracy when the unlabeled data is drawn from the same distribution as the limited set of labeled data. This is the classic benchmark to evaluate SEMI-SL algorithms, however, it can readily be applied to SELF-SL methods as well. Even though the SELF-SL methods' performance have been improving drastically

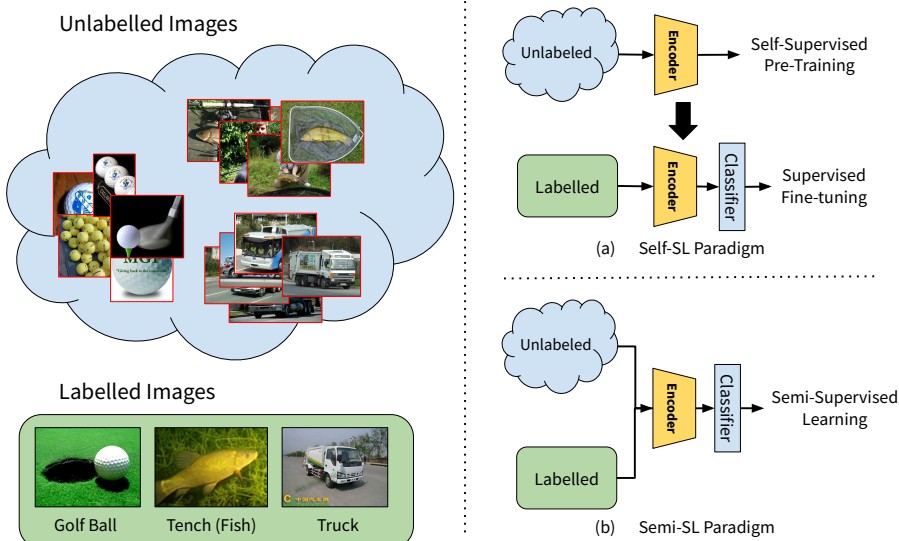

Figure 1: Limited Label Representation Learning is the task of learning a strong classifier using a small amount of labelled data along with a large amount of unlabeled data. Two popular paradigms for solving this task are **(a)** Self-Supervised pre-training on the unlabeled data followed by finetuning on the labelled data and **(b)** Semi-Supervised learning, where labeled and unlabeled data are both used together during joint representation and classifier learning.

in the past few years, our analysis suggests that SEMI-SL methods outperform SELF-SL methods even when given access to a very limited annotated set. Unsurprisingly, this performance gap narrows as the size of the annotated data grows.

Our next set of experiments concern the robustness of the learned representations. The first robustness criterion that we examine is *out-of-distribution (OOD) detection* Hendrycks & Gimpel (2017). Particularly, we evaluate predictive confidence of classifiers based on SEMI-SL and SELF-SL against data that is drawn from a different semantic space than training data. A practical classifier should produce low-confidence predictions against OOD data. Utilizing a small set of annotated data, our experiments indicate SEMI-SL surpasses SELF-SL in OOD detection. The OOD detection is vital for deploying models that encounter a significant distribution shift at test time. To deploy models in safety-critical applications Ghaffari Laleh et al. (2022), another quality that is essential is robustness against corruption and adversarial attacks. Therefore, subsequently, we evaluate *adversarial robustness* and *robustness to corruption* of the SEMI-SL and SELF-SL models, where SEMI-SL outperforms SELF-SL on both benchmarks.

A key goal of representation learning from large datasets is to reuse representations across multiple downstream tasks. We evaluate the transferability of SEMI-SL and SELF-SL learned features to near-domain image classification through retrieval Krause et al. (2013); Fei-Fei et al. (2004); Bossard et al. (2014); Berg et al. (2014); Parkhi et al. (2012); Nilsback & Zisserman (2008) and fine-grained Radenović et al. (2018) image retrieval tasks. Utilizing retrieval allow us to test the general utility of the learned features themselves without finetuning. The chosen classification datasets are considered *near-domain* as they share significant overlap with the ImageNet dataset used for pre-training. We also study transferability in cross-domain settings such as satellite images and medical images with few-shot training examples.

Finally, we also compare the robustness of SELF-SL and SEMI-SL methods to class imbalance in the training data. Since imbalance is a natural property of most real world sources of unlabeled data, this is a crucial test for limited label representation learning methods.

To briefly summarize, the key contributions of our work are as follows:

- An extensive, fair and systematic evaluation of comparable SELF-SL and SEMI-SL methods,
- Demonstration of robustness and transferability benefits of SEMI-SL over SELF-SL,
- Establishing the importance of early introduction of labels in limited label representation learning.

## 2 RELATED WORK

### 2.1 SELF-SUPERVISED LEARNING

In recent years, many classes of self-supervised learning methods have become popular for limited label representation learning. The standard frameworks follow a two stage approach: the first stage is self-supervised representation learning followed by supervised finetuning on the labelled set. Popular families of SELF-SL methods include Contrastive learning based, Clustering based, Self Distillation based and Masked Image Modeling based methods.

**Contrastive Learning** methods such as SimCLR Chen et al. (2020a) carry out representation learning by forming positive pairs from image datasets by applying semantics preserving data augmentation and using other data instances to form negative pairs. Some works build upon the simple contrastive learning framework by complementing it with techniques such as Momentum Contrast (MoCo family He et al. (2020); Chen et al. (2020b; 2021)) and Nearest Neighbour hard positive mining (NNCLR Dwibedi et al. (2021)).

**Clustering Based** SELF-SL: DeepCluster Caron et al. (2018) learns an image representation by alternatively carrying out clustering using the learned features and learning features by predicting the cluster assignment from the previous clustering step. SwAV Caron et al. (2020) takes clustering online using the Sinkhorn-Knopp algorithm and combined the idea of swapping the cluster assignment to be predicted between multiple augmented views of the same image.

**Self-Distillation Based** SELF-SL: Methods such as BYOL Grill et al. (2020) and DINO Caron et al. (2021b) maintain an exponential moving average of the model being trained as a teacher and train the student model to match the teacher's prediction. In order to prevent collapse an additional predictor head and feature centering are used.

**Masked Image Modeling Based** SELF-SL: MIM based methods focus on learning representations by reconstruction. MAE He et al. (2022) uses an encoder-decoder based approach, with a lightweight decoder for pixel reconstruction. MaskFeat Wei et al. (2022) utilizes a simple linear layer on top of a vision transformer encoder for reconstructing low level features of the masked area.

### 2.2 SEMI-SUPERVISED LEARNING

Many approaches for semi-supervised learning have been explored in the literature: combining a self-supervised loss on the unlabeled images with a supervised loss on the labeled images, imposing consistency regularization across multiple predictions from the same unlabeled image, self-training using psuedo-labels, and hybrid approaches which combine these techniques.

S4L Zhai et al. (2019) combines the self-supervised rotation prediction pretext task along with supervised loss on labeled images. Consistency regularization based approaches include Π-Model Laine & Aila (2017) which imposes consistency between different network outputs obtained using dropout. Mean Teacher Tarvainen & Valpola (2017) uses consistency between a "student" model and an exponential moving average "teacher" version of itself. UDA Xie et al. (2020) simply generates two views of the input data using data augmentation for consistency training. PAWS Assran et al. (2021) utilizes samples of the labeled data as a support set for predicting view assignments for unlabeled images, and utilizes consistency between view assignments of augmented views of a single image for training. Psuedo-labeling Lee et al. (2013) the unlabeled data using a model trained on the labeled data and then iteratively self-training on the generated labels is a simple and effective method for Semi-SL. UPS Rizve et al. (2021) proposed an improved method for assigning psuedo-labels using label uncertainty estimation. Hybrid methods such as MixMatch Berthelot et al. (2019b) and FixMatch Sohn et al. (2020) combine elements of both psuedo-labeling and consistency regularization. Some hybrid methods such as ReMixMatch Berthelot et al. (2019a) also combine the previously mentioned techniques with imposition of a prior on the label distribution of the unlabeled images. SimMatch Zheng et al. (2022) uses a dual-consistency approach, where each unlabeled instance is assigned an "instance" psuedo-label based on its similarity with labeled instances, and a "semantic" psuedo-label based on its distance to class centers. Both consistency loss and self-training on psuedo-labels are used, which makes SimMatch one of the most powerful Semi-SL techniques.

## 3 SEMI AND SELF SUPERVISED LEARNING

In this section we revisit the SELF-SL and SEMI-SL as the two dominant approaches to deal with the challenge of training with a limited labeled dataset $X_l$ through an auxiliary unlabeled dataset $X_u$. Suppose $X_l = \{(x_i, y_i) \in X \times Y\}_{i=1..n}$, where $X \subset R^r$ denotes the input space and $Y \subset R^K$ denotes the classification probability simplex for $K$ classes. Moreover, consider $X_u = \{u_i \in U\}_{i=1..m}$, where $U \subset R^r$ denotes the set of unlabeled data. Our goal is to learn $h \circ f$ that models $p(y|x)$, where $f_\theta$ is the network that learns the representation (feature extractor) and $h_\phi$ is the classification head. The main challenge in learning SEMI-SL and SELF-SL is to learn representations from unlabeled examples that enhance generalization of the classifier. To this end, both approaches rely on some assumptions about the data distribution.

One common assumption about *data distribution* is data consistency, which presumes small variations of a data sample should not produce large variations in its representation. In other words, $f(x) \approx f(x')$ given $x' = T(x); T \in \mathcal{T}$, where $\mathcal{T}$ is a set of valid transformations such as the data augmentations that do not change the classification label of an image. However, data consistency can lead to the degenerate solution, where representations of different samples collapse; therefore various samples are assigned to an identical representation. To resolve such a mode collapse, data consistency is usually applied in conjunction with *clustering assumption*, where, the data distribution undertakes at least $K$ ($K$ is the number of the clusters) distinct modes. From this perspective, the training objective for unlabeled data is defined as:

$$loss_u = \underbrace{\sum_i sim_g(T(u_i), T'(u_i))}_{\text{consistency obj.}} - \lambda \underbrace{\sum_{u_i \in c_i, u_j \notin c_i} sim_g(T(u_i), T'(u_j))}_{\text{clustering obj.}}, \qquad (1)$$

where $c_i$ is the $i$th cluster and $sim_g(.,.)$ is a function that measures similarity of two samples in a representation space induced by a function $g()$[1].

Most of the successful SEMI-SL and SELF-SL methods such as pseudo-labeling (Lee et al., 2013; Rizve et al., 2021), FixMatch (Sohn et al., 2020), contrastive learning (Chen et al., 2020a; Tian et al., 2020), and Dino (Caron et al., 2021a) are based on data consistency and clustering assumption. Therefore, these methods follow Eq.1. For example, in contrastive learning methods such as SimCLR (Chen et al., 2020a), every sample defines a cluster and the training objective is to minimize the negative log-likelihood:

$$\sum_i \log[\sum_{u_k \notin c_i} \exp(cos\_sim(f(T(u_i)), f(T'(u_k)))/\tau)] - \log \exp(cos\_sim(f(T(u_i)), f(T'(u_i)))/\tau),$$
$$(2)$$

where $cos\_sim()$ denotes cosine similarity function and $\tau$ is the temperature scaling parameter.

A main difference between SEMI-SL and SELF-SL is in the timeline for introducing the labeled data. Particularly, SEMI-SL methods introduce these examples early during the training by including a supervised learning loss such as cross entropy. However, SELF-SL methods uncouple representation learning and the classification stage. As a result, a common postulate is that SELF-SL methods are better suited to learn more generalizable representations, i.e., more fitted for transfer learning, than Semil-SL algorithms. We study the transferability property of the two approaches in Section 4.3, where we find that across all tasks, SEMI-SL methods match or outperform SELF-SL methods.

Another major disparity is in the definition of the clusters. While in semi-supervised learning the dominant approach is assigning a cluster to each class, in contrastive learning every sample represents a cluster. Furthermore, there are SELF-SL methods, such as Dino, in which the number of clusters is a hyperparameter. The mismatch between the number of clusters and the training examples leads to various approaches to assign examples to the clusters. A successful technique in SEMI-SL is to dynamically assign samples to clusters following their predictive confidence (Sohn et al., 2020). Alternatively, Dino applies centering to dynamically assign examples to the predefined clusters.

To compare SEMI-SL and SELF-SL approaches, we adopt two representative methods from SEMI-SL and their matching counterparts from SELF-SL. In particular, we compare SimMatch and PAWS with SimCLR and SwAV, respectively. Figure 2 shows a schematic overview of these methods. In the following we briefly review them.

---

[1]Common choices for $g$ are $f$ and $h \circ f$.

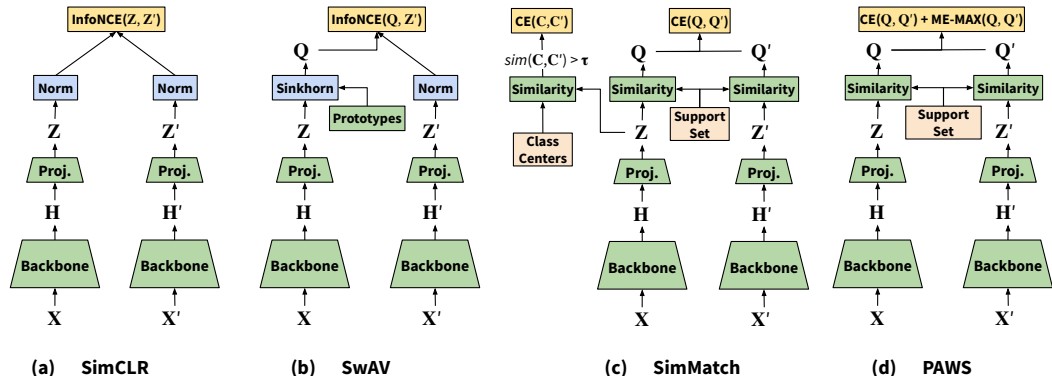

Figure 2: Self and Semi-Supervised Learning Methods analyzed in this work. **ME-MAX** in PAWS denotes mean entropy maximization which maximizes the average of the sharpened predictions.

**SimCLR:** One of the earliest approaches in SELF-SL through contrastive learning is SimCLR. As presented in Figure 2, SimCLR utilizes two semantic preserving transformations of an image to generate members of a positive pair that are trained to produce consistent representations. Whereas it regards the other images from a batch as negative samples that belong to other clusters and, therefore, their representations are dissociated. The training objective is called InfoNCE, which combines these consistency and clustering objectives according to Equ. 2.

**SwAV:** Another prominent method in SELF-SL is contrasting cluster assignments across views (SwAV), which also utilizes the paired augmentation strategy of SimCLR to generate matching views for unlabeled data to impose data consistency. However, SwAV promotes this consistency between cluster assignments of different views of the same image, contrary to SimCLR which implement the consistency directly on image features. In particular, SwAV replaces the similarity between two representation vectors $sim(g(u), g(u'))$ by $sim(g(u), c_{u'}) + sim(g(u'), c_u)$, where $u' = T'(u)$ and $c_u$ denotes the soft cluster assignment of $u$. On top of that, SwAV utilizes Sinkhorn-Knopp algorithm to guarantee that the cluster assignments of examples is uniform. In addition, SwAV introduces a set of prototype vectors to ameliorate the necessity of large batch sizes that are applied in contrastive learning methods in order to increase the variety of the clusters (negative pairs).

**PAWS:** Predicting view assignments with support samples (PAWS) is a SEMI-SL method that takes advantage of the set of labelled samples, called support set, instead of learning a set of prototypes as in SwAV. This method encourages the similarity between the two views of the same image by minimizing the cross-entropy between their pseudo-labels (akin to soft cluster assignments in Eq. 1). The pseudo-labels are generated by a soft nearest neighbor classifier that makes a weighted average of the labels in the support set according to the distance of the unlabeled sample to the samples in the support set: $\sum_{x_j \in X, y_j \in Y} \frac{d(f(u_i), f(x_j))}{\sum_{x_k \in X} d(f(u_i), f(x_k))} y_j$, where $d(.,.)$ is a distance function. Finally, to avoid representation collapse, PAWS sharpens the pseudo-labels and applies a mean entropy maximization term that promotes assignment to different labelled samples.

**SimMatch:** Another prominent SEMI-SL method is SimMatch, which proposes to improve consistency of the learned representation by a semantic level-similarity and an instance-level similarity. Following FixMAtch, the semantic similarity minimizes the cross-entropy loss between pseudo-labels of a weakly augmented and a strongly augmented view of an image. This loss is only applied to the samples with a high predictive confidence of their weakly augmented view. The instance-level similarity first convert the weak view and strong view representations to a distribution by utilizing their distances to the representations of the labeled samples. Then a cross-entropy loss promotes the agreement of these two distributions. Note that the labeled samples are simultaneously trained with a supervised learning loss.

## 4 ANALYSIS

In this section we compare different aspects of the representations learned by SEMI-SL and SELF-SL approaches. Our analysis is organized into four sets of experiments: in-domain performance, robustness, transferability and the ability to learn from imbalanced data.

### 4.1 IN-DOMAIN PERFORMANCE

In-Domain performance is the classic framework to evaluate SEMI-SL algorithms. In this setting, both the unlabelled and labelled data used for representation learning are from the same domain (in this case ImageNet). The downstream task is the classification of the same domain (ImageNet) as well.

Intuitively, in this task, we expect SEMI-SL to outperform SELF-SL, as features relevant for the downstream task can be learned from the very beginning of representation learning in SEMI-SL, whereas SELF-SL does not have access to labels and hence the learned representation is not tailored to the given set of classes. This matches with the empirical observations in Table 4.1, where in the 1% labelled dataset setting SEMI-SL methods (PAWS and SimMatch) outperform SELF-SL techniques by 7%. As the amount of labelled data increases (10%), the gap between SEMI-SL and SELF-SL closes and we observe that PAWS outperforms SwAV only by <1%.

| Method | ImageNet Top-1 Accuracy | |
| --- | --- | --- |
| Labelled data → | 1% | 10% |
| *Semi-Supervised* | | |
| **PAWS** | 66.48 | **75.43** |
| **SimMatch** | **67.11** | 74.15 |
| *Self-Supervised* | | |
| **SwAV** | 59.34 | 74.99 |
| **SimCLR** | 45.95* | 65.35 |

Table 1: In-Domain Classification Performance. * - our reproduction

### 4.2 ROBUSTNESS

#### 4.2.1 OUT-OF-DOMAIN DETECTION

In our first task, we evaluate robustness of SELF-SL and SEMI-SL approaches to out-of-domain (OOD) detection . In these experiments, an OOD dataset is created by sampling images from LSUN-Scenes and ImageNet21k datasets, respectively, where we remove the classes that are in common with ImageNet1k. The ImageNet1K Validation set is used as the in-domain dataset. The finetuned ImageNet classifiers are used as OOD detectors by treating the softmax score of the highest predicted class as the predicted in-domain score. Following prior work (Hendrycks & Gimpel, 2016; Liang et al., 2017), AUPR and AUROC metrics are calculated for the binary task of OOD detection. When 1% labelled data is available, SEMI-SL outperforms SELF-SL. Nevertheless, with the availability of more labelled data, in the 10% setting, both classes of methods perform similarly with small difference in AUPR and AUROC. As OOD detection can be important for practical deployment of models, the choice of SEMI-SL vs SELF-SL can make a difference.

| OOD Dataset → | LSUN-Scenes | | | | ImageNet21K − ImageNet1K | | | |
| --- | --- | --- | --- | --- | --- | --- | --- | --- |
| Labelled data → | 1% | | 10% | | 1% | | 10% | |
| **Method** | *AUPR* | *AUROC* | *AUPR* | *AUROC* | *AUPR* | *AUROC* | *AUPR* | *AUROC* |
| *Semi-Supervised* | | | | | | | | |
| **PAWS** | 0.689 | 0.659 | 0.742 | 0.735 | 0.687 | 0.710 | 0.713 | 0.737 |
| **SimMatch** | **0.756** | **0.732** | 0.734 | 0.732 | **0.711** | **0.726** | **0.716** | **0.741** |
| *Self-Supervised* | | | | | | | | |
| **SwAV** | 0.649 | 0.617 | **0.752** | **0.745** | 0.658 | 0.659 | 0.715 | 0.727 |
| **SimCLR** | 0.606 | 0.571 | 0.699 | 0.678 | 0.625 | 0.618 | 0.662 | 0.675 |

Table 2: Out-of-Domain Detection

#### 4.2.2 ADVERSARIAL ROBUSTNESS

The adversarial robustness is measured by using popular adversarial attacks such as FGSM and PGD. Images from the ImageNet validation set are used for generating the adversarial examples. This setting highlights a major advantage of SEMI-SL methods over SELF-SL as in both 1% and

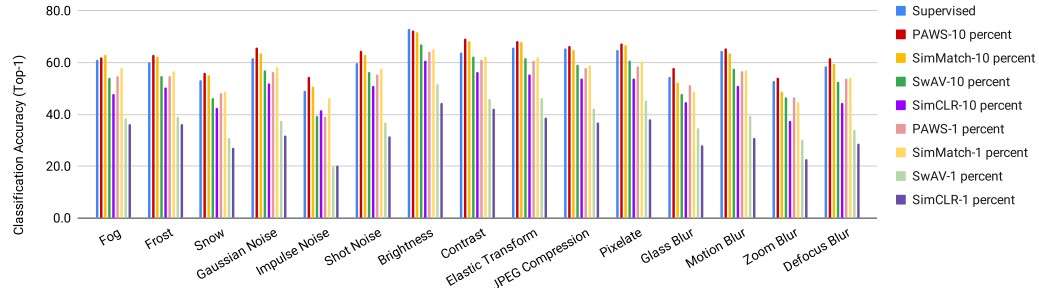

Figure 3: Effect of different types of corruptions

10% settings SEMI-SL outperforms SELF-SL by large a margin. As reported in Table 3, in the 10% labelled data setting, PAWS outperforms SwAV by 6% and 8% under FGSM and PGD attacks, respectively, while SimMatch outperforms SimCLR by even a larger margin. We hypothesize that the availability of labels during representation learning allows SEMI-SL methods to learn robust representations that lead to maintaining larger margins between class boundaries and class instances. In applications where robustness is important, SEMI-SL representations therefore provide a clear advantage over SELF-SL models.

| Method | 1% labelled data | | | 10% labelled data | | |
|---|---|---|---|---|---|---|
| | Clean Acc.↑ | Robust Acc. ↑ | | Clean Acc.↑ | Robust Acc. ↑ | |
| | | *FGSM* $\epsilon = 1/255$ | *PGD* $L2\ \epsilon = 0.5$ | | *FGSM* $\epsilon = 1/255$ | *PGD* $L2\ \epsilon = 0.5$ |
| *Semi-Supervised* | | | | | | |
| **PAWS** | 66.48 | 17.56 | 19.01 | 75.43 | 14.90 | 15.79 |
| **SimMatch** | 67.11 | **25.94** | **25.70** | 74.15 | **26.59** | **26.91** |
| *Self-Supervised* | | | | | | |
| **SwAV** | 59.34 | 6.25 | 6.43 | 74.99 | 8.91 | 7.28 |
| **SimCLR** | 45.95 | 10.37 | 9.29 | 65.35 | 11.67 | 9.65 |

Table 3: Adversarial Robustness

### 4.2.3 ROBUSTNESS TO NATURAL CORRUPTIONS

Adversarial robustness represents the worst case perturbation that a classifier could encounter, however, in practice robustness to natural corruptions such as blurring and fog is more important. To measure robustness to natural corruptions we utilize ImageNet-C dataset Hendrycks & Dietterich (2019) and measure mean Corruption Error (mCE) across 15 different perturbation types relative to a supervised ResNet50 model, which is assigned an mCE score of 100. Lower values indicate higher robustness. In this experiment, SEMI-SL methods outperform SELF-SL methods by nearly $\sim 40$ points in the

| Method | Mean Corruption Error (mCE1) ↓ | |
|---|---|---|
| | *labelled data* | |
| | *1%* | *10%* |
| *Semi-Supervised* | | |
| **PAWS** | 115.4 | **93.4** |
| **SimMatch** | **112.2** | 97.7 |
| *Self-Supervised* | | |
| **SwAV** | 151.2 | 109.5 |
| **SimCLR** | 171.7 | 128.6 |

Table 4: Robustness to Natural Corruptions

1% setting and $\sim 10$ points in the 10% setting (Table 4). This experiment confirms that the labels are crucial for learning robust features. Figure 3 illustrates the effect of each type of corruption.

### 4.3 TRANSFERABILITY

Transfer learning is a common practical application of representation learning. Recent progress in SELF-SL methods show their superior transferability over classic supervised pretraining approach. In this section we compare the transfer learning capabilities of SELF-SL and SEMI-SL methods.

### 4.3.1 NEAR-DOMAIN TRANSFERABILITY

In order to evaluate the transferability of the features learned by SEMI-SL and SELF-SL methods in a near-domain setting, we formulate a simple retrieval based classification task. We do not fine-tune

the models in order to evaluate the natural transferability of the learned features. Alternatively, a k-nearest neighbour classifier is built using the training set of the evaluated problem, and the images from the val/test are classified by matching with the training set using the feature representation of the model being evaluated.

| | BirdSnap | Cars | Food | Flowers | CalTech | Pets | Average |
|---|---|---|---|---|---|---|---|
| *Semi-Supervised* | | | | | | | |
| **SimMatch** (1%) | 35.7 | 23.4 | 49.6 | 81.6 | 88.8 | 87.9 | 55.8 |
| **SimMatch** (10%) | 36.6 | 30.2 | 51.4 | 82.5 | **89.1** | **89.1** | 57.9 |
| **PAWS** (1%) | 34.0 | 32.8 | 53.4 | **87.3** | 87.0 | 84.5 | 58.9 |
| **PAWS** (10%) | **37.8** | **35.9** | 55.0 | 87.2 | 88.4 | 85.9 | **60.9** |
| | | | | | | | |
| *Self-Supervised* | | | | | | | |
| **SimCLR** | 12.6 | 16.0 | 39.6 | 77.2 | 85.5 | 68.3 | 46.2 |
| **SimCLR** (1% FT) | 15.6 | 14.6 | 39.6 | 74.2 | 84.6 | 71.3 | 50.0 |
| **SimCLR** (10% FT) | 22.0 | 18.9 | 43.7 | 75.8 | 87.4 | 79.2 | 49.6 |
| **SwAV** | 16.4 | 21.6 | 49.9 | 83.0 | 84.7 | 75.3 | 51.1 |
| **SwAV** (1% FT) | 26.5 | 24.4 | 52.0 | 83.6 | 83.9 | 78.9 | 54.1 |
| **SwAV** (10% FT) | 34.5 | 27.2 | **55.5** | 85.7 | 87.0 | 84.4 | 58.0 |

Table 5: Near-Domain Transferability. Best overall & 1% result **Bolded** & Underlined.

Note that all of these datasets consist of near-domain examples from ImageNet, where the images have label overlap with ImageNet. Moreover, CalTech and Pets datasets have a high degree of overlap with ImageNet classes, while BirdSnap and Cars have limited overlap, i.e., representing a harder generalization task. As a result, we expect higher accuracies on CalTech and Pets.

We find that across all tasks, SEMI-SL methods match or outperform SELF-SL methods in transferability (Table 5). At the first glance, transferability is one of the tasks in which we expect SELF-SL to have an advantage over SEMI-SL methods as their representation learning is not tied to one set of labels. However, as we observe in this Table, even in transferability, SEMI-SL methods maintain their advantage. This suggests that the semantic information in the source domain improves the quality of learned features for near-domain tasks.

### 4.3.2 FINE-GRAINED RETRIEVAL

This analysis is siilar to the near-domain transferability, however it utilizes a fine-grained task. In this experiment, the images are from the revisited Oxford and Paris datasets, where one specific building forms a single "*class*". The result in Table 6 show that SEMI-SL methods outperform SELF-SL by significant margin across Easy, Hard and Medium settings. Therefore, despite being trained on coarse grained class labels, SEMI-SL methods still benefit on this fine-grained task.

| | rOxford5k (mAP) | | | rParis6k (mAP) | | |
|---|---|---|---|---|---|---|
| **Method** | **Easy** | **Medium** | **Hard** | **Easy** | **Medium** | **Hard** |
| *Semi-Supervised* | | | | | | |
| **SimMatch** (1%) | 50.33 | 33.57 | 7.03 | 76.34 | 60.72 | 32.46 |
| **SimMatch** (10%) | 48.84 | 33.38 | 8.05 | 74.47 | 59.29 | 31.80 |
| **PAWS** (1%) | 57.61 | 38.79 | 8.93 | 75.33 | 59.73 | 32.44 |
| **PAWS** (10%) | **54.98** | **38.07** | **10.36** | **78.12** | **61.39** | **33.70** |
| *Self-Supervised* | | | | | | |
| **SimCLR** | 34.92 | 23.56 | 4.26 | 66.37 | 50.07 | 22.76 |
| **SimCLR** (1% FT) | 38.77 | 26.27 | 4.63 | 68.51 | 52.80 | 25.01 |
| **SimCLR** (10% FT) | 39.80 | 26.49 | 4.67 | 70.84 | 53.95 | 25.49 |
| **SwAV** | 47.15 | 32.25 | 7.94 | 71.91 | 53.22 | 24.08 |
| **SwAV** (1% FT) | 49.86 | 32.58 | 6.06 | 73.89 | 56.13 | 26.99 |
| **SwAV** (10% FT) | 52.56 | 34.82 | 7.54 | 75.02 | 57.86 | 29.28 |

Table 6: Fine-Grained Retrieval. Best overall & 1% result **Bolded** & Underlined.

### 4.3.3 CROSS-DOMAIN FEW-SHOT TRANSFERABILITY

To compare the transferability of learned features by these methods to completely different data domains, we evaluate their few-shot classification performance in the cross-domain setting . The datasets Guo et al. (2020) on which we evaluate the few-shot performance exhibits various shifts from in-domain ImageNet data. For instance, even though the Crop Disease dataset contains natural images, it covers drastically different semantic concepts that are specific to the agricultural industry. EuroSAT, on the other hand, not only contains semantically different images but also includes images without any perspective distortion. ISIC and ChestX datasets pose greater challenge since both of these datasets contain medical images, with ChestX images lacking even color.

To minimize the impact of confounding variables in this evaluation we do not perform any finetuning. We use the pretrained models from the corresponding methods to extract features and learn a logistic regression classifier on top of the extracted features by utilizing the few labeled examples from the support set. We report the results in Table 7, where, we observe that, on ISIC and EuroSAT datasets, the learned representations from PAWS outperforms SwAV, whereas, on Crop Disease and ChestX datasets the difference is smaller than the margin of error. We observe a similar trend in the case of SimCLR and SimMatch where SimMatch outperforms SimCLR on all datasets except ChestX, where it ties with SimCLR. These results demonstrate that, even on drastically different data domains, transferability of SEMI-SL features either outperforms or matches SELF-SL methods.

| | Crop Disease | | EuroSAT | | ISIC | | ChestX | |
|---|---|---|---|---|---|---|---|---|
| | 1-shot | 5-shot | 1-shot | 5-shot | 1-shot | 5-shot | 1-shot | 5-shot |
| *Semi-Supervised* | | | | | | | | |
| **SimMatch** (1%) | 74.95 | 93.30 | 68.36 | 87.04 | 30.82 | 43.91 | 22.90 | 26.56 |
| **SimMatch** (10%) | 76.15 | 93.80 | 69.28 | 87.60 | 30.51 | 44.37 | 22.46 | 25.81 |
| **PAWS** (1%) | 79.47 | 94.92 | **71.55** | 89.77 | 32.03 | **46.79** | 23.27 | 27.12 |
| **PAWS** (10%) | 80.32 | 95.16 | 70.04 | **89.96** | **31.72** | 45.82 | **23.33** | 27.23 |
| *Self-Supervised* | | | | | | | | |
| **SimCLR** | 73.11 | 92.60 | 61.35 | 84.73 | 30.74 | 43.97 | 23.51 | **27.90** |
| **SimCLR** (1% FT) | 72.04 | 91.54 | 61.55 | 79.91 | 29.12 | 40.55 | 22.65 | 25.88 |
| **SimCLR** (10% FT) | 71.34 | 92.27 | 63.46 | 84.63 | 29.54 | 42.92 | 22.94 | 26.93 |
| **SwAV** | 80.02 | 95.07 | 69.99 | 89.27 | 29.25 | 42.59 | 22.98 | 26.53 |
| **SwAV** (1% FT) | 80.06 | 95.36 | 68.97 | 87.67 | 30.05 | 43.26 | 22.81 | 26.36 |
| **SwAV** (10% FT) | **80.60** | **95.59** | 68.53 | 88.47 | 29.71 | 43.80 | 22.52 | 26.00 |

Table 7: Cross-Domain Few Shot Transfer Learning. Best overall & 1% result **Bolded** & Underlined.

### 4.4 LEARNING FROM IMBALANCED DATA

To compare SEMI-SL and SELF-SL on their ability to learn from imbalanced data we utilize subsets of ImageNet-100 for reasonable training time within our computational budget. For training, we create two subsets of ImageNet-100, each with 58,200 unlabelled images and 6,500 labelled images. In the balanced subset, equal number of images from each of the 100 classes are included in both labelled and unlabeled groups. Whereas, in the Imbalanced subset, the number of im-

| Top-1 Accuracy on ImageNet-100 | | |
|---|---|---|
| | **SwAV** | **PAWS** |
| **Balanced** | 36.7 | 48.3 |
| **Imbalanced** | 24.4 | 47.4 |

Table 8: Learning from Imbalanced Data.

ages from each class match a distribution with Imbalance Factor of 5. The entire ImageNet-100 validation set is used for testing. Both SwAV and PAWS models are trained for 100 epochs. The results in Table 8 demonstrate that SEMI-SL (PAWS) outperforms SELF-SL (SwAV) in both balanced and imbalanced settings. It is also noteworthy that SEMI-SL is less sensitive to imbalance in data.

## 5 CONCLUSIONS

As self-supervised pre-training followed by finetuning becomes the dominant paradigm for limited label representation learning, this study provides evidence for the continued utility of SEMI-SL. In particular, our detailed empirical studies demonstrate that early introduction of labels in SEMI-SL benefits both the robustness and transferability of the learned representation. Moreover, our analysis illustrates that SEMI-SL leads to representations that are more resilience to imbalance in data.

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

# A APPENDIX

## A.1 EXPERIMENTAL DETAILS

In our experiments, we obtain the analyzed models from the repositories of the original authors.

The links to each of these are provided below:

**SwAV**: `https://github.com/facebookresearch/swav`

**SimCLR**:`https://github.com/google-research/simclr`

**PAWS**:`https://github.com/facebookresearch/suncet`

**SimMatch**:`https://github.com/mingkai-zheng/SimMatch.git`

Since the SimCLR repository does not provide weights for the 1% finetuned model, we train such a model by finetuning it on 1% of data. The finetuning job ran for 20 epochs, with a learning rate of 5.0 for the classification head and 0.2 for the backbone. LR decay of 0.2 was applied at the 12th and 16th epochs. A Batch Size of 256 (128 across 2 RTX A6000 GPUs with 48GB VRAM) was used for the finetuning.

### A.1.1 OOD DETECTION

In our OOD experiments, we use a temperature of 1.0 for calculating the softmax score.

### A.1.2 ADVERSARIAL ROBUSTNESS

For the PGD attack we use a $L_2$ norm magnitude of 0.5, with 4 attack steps.

### A.1.3 ROBUSTNESS TO NATURAL CORRUPTIONS

The original ImageNet-C dataset has 5 severity levels of corruption, we only use severity level 1 since the models under test are not especially trained for robustness and severity = 1 causes a significant drop in robustness.

Unlike the original paper which uses an AlexNet model as reference for calculating the relative mean corruption error, we use a Supervised ResNet50 model as the reference as all the models we are testing are ResNet50 and this shifts the scale to a natural reference point of 100.

### A.1.4 NEAR-DOMAIN TRANSFERABILITY

For the k-nearest neighbour classifier we use $k$=1. In our experiments different values of $k$ do not change the relative ordering of the models.

### A.1.5 FINE-GRAINED RETRIEVAL

We use the Revisited Oxford and Paris images without distractors for these experiments.

## A.2 EVALUATING REPRESENTATIONS IN DENSE ASSOCIATION TASKS

We use the UniTrack Wang et al. (2021) codebase for these experiments:

`https://github.com/Zhongdao/UniTrack`

### A.2.1 VIDEO OBJECT SEGMENTATION

We evaluate the model representation on DAVIS-2017 Video Object Segmentation task following the protocol in Jabri et al. Jabri et al. (2020) The segmentation mask is propagated by nearest neighbour matching between consecutive frames without any trainable component. This provides us an evaluation of the spatial representation capabilties of the models. We report mean region similarity ($\mathcal{J}_m$) and contour accuracy ($\mathcal{F}_m$) relative to the ground truth segmentation. Region similarity is measured using Intersection over Union, while contour accuracy is measured using F-measure, which

is a weighted average of the recall and precision. We find that SEMI-SL methods slightly outperform SELF-SL models on this task.

| | $(\mathcal{J}\&\mathcal{F})_m$ | $\mathcal{J}_m$ | $\mathcal{F}_m$ |
|---|---|---|---|
| *Semi-Supervised* | | | |
| **PAWS** (1%) | 62.2 | 60.6 | 63.8 |
| **PAWS** (10%) | 62.0 | 60.2 | 63.9 |
| **SimMatch** (1%) | 63.4 | 61.3 | 65.4 |
| **SimMatch** (10%) | **63.6** | **61.6** | **65.6** |
| *Self-Supervised* | | | |
| **SimCLR** | 62.3 | 60.5 | 64.2 |
| **SimCLR** (1% Finetuned) | 58.1 | 56.5 | 59.7 |
| **SimCLR** (10% Finetuned) | 61.6 | 59.7 | 63.5 |
| **SwAV** | 62.7 | 61.1 | 64.3 |
| **SwAV** (1% Finetuned) | 62.2 | 60.5 | 63.9 |
| **SwAV** (10% Finetuned) | 61.9 | 60.2 | 63.7 |

Table 9: Video Object Segmentation

### A.2.2 MULTI-OBJECT TRACKING

We follow the UniTrack framework Wang et al. (2021) for this evaluation. The ground truth detections are provided to the framework, and data association across frames is carried out with features extracted using the model being tested. SEMI-SL methods outperform SELF-SL by around $1.5\%$ in IDF1 (which measures association accuracy) and around $1\%$ in the composite HOTA metric.

| | IDF1 | HOTA |
|---|---|---|
| Semi-Supervised | | |
| **PAWS** (1%) | 77.6 | 64.1 |
| **PAWS** (10%) | 78.0 | 64.5 |
| **SimMatch** (1%) | 78.1 | 64.5 |
| **SimMatch** (10%) | **78.8** | **64.8** |
| Self-Supervised | | |
| **SimCLR** | 76.1 | 63.2 |
| **SimCLR** (1% Finetuned) | 77.2 | 63.5 |
| **SimCLR** (10% Finetuned) | 76.9 | 63.5 |
| **SwAV** | 70.9 | 59.5 |
| **SwAV** (1% Finetuned) | 77.3 | 63.7 |
| **SwAV** (10% Finetuned) | 77.2 | 63.9 |

Table 10: Multi-Object Tracking

