# OpenReview forum: "Self-Supervision is Not All You Need: In Defense of Semi-Supervised Learning"
_ICLR.cc/2024/Conference — Submitted to ICLR 2024_

### Official Review · Reviewer_1DSs · 2023-10-15

**Soundness:** 2 fair
**Presentation:** 3 good
**Contribution:** 2 fair
**Rating:** 5
**Confidence:** 4

**Summary:**

This paper provides comprehensive empirical studies in comparing self-supervised learning (SELF-SL) and semi-supervised learning (SEMI-SL). Specifically, this paper tries to answer the question “When to favor one over the other?”. In particular, the authors investigates the cases of in-domain, near-domain and out-of-distribution data, robustness to image corruptions and adversarial attacks, cross-domain few-shot learning, and the ability to learn from imbalanced data.

**Strengths:**

1.	The paper is generally written in a clear way.
2.	The experiments are sufficient.

**Weaknesses:**

1.	Some claims in this paper are not proper. For example, the authors claim that “These two lines of research (i.e., SELF-SL and SEMI-SL) have progressed largely independent of each other”. However, to my best knowledge, many semi-supervised learning borrows the idea of contrastive learning, which is an important SELF-SL paradigm, e.g., CoMatch: Semi-supervised Learning with Contrastive Graph Regularization (ICCV 21), CLDA: Contrastive Learning for Semi-Supervised Domain Adaptation (NIPS 21), Class-Aware Contrastive Semi-Supervised Learning (CVPR 22), etc.
2.	I’m not sure whether it is fair to directly compare SELF-SL and SEMI-SL. Firstly, it should be noted that in many practical cases, we can easily acquire the pre-trained large model from web, but the abundant data for training such large model are not released due to privacy issue (except the models pre-trained on ImageNet or other public datasets). When we come to the present problem, what we have are small amount of data-of-interests at hand and the available pre-trained model. At this time, we can only use SELF-SL. Secondly, even we have the training data (maybe unlabeled) for pre-trained model, the distribution of these data is different from that (maybe labeled) in the current task. SELF-SL perfectly fits this setting. However, for SEMI-SL, the labeled data and unlabeled data are usually generated from the same distribution, but SELF-SL will probably fail in this setting. Therefore, I think in many cases, SELF-SL and SEMI-SL are not directly comparable, as they have their own advantages in specific setting and application.
3.	The number of tables in this paper is strange. The authors mentioned that “This matches with the empirical observations in Table 4.1.” However, I cannot find Table 4.1 at all.
4.	This paper conducts comprehensive experiments in comparing SELF-SL and SEMI-SL, which is good. However, it would be better if more analyses can be provided for the results in different investigated tasks.

**Questions:**

See my comments on weaknesses.

---

> ### Author Response · Authors · 2023-11-23
>
> 1. We agree that some methods have tried combining SemiSL and SelfSL training paradigms, we do discuss some such methods in our related work sections. However, most works in either SemiSL or SelfSL limited learning settings avoid comparative analyses with the other category.
>
> 2. We agree that there exist practical settings where SemiSL and SelfSL methods are not equivalent. However, the setting outlined in the paper (i.e. availability of large amounts of unlabelled data and small amount of labels)  is also frequently encountered by practitioners, and of broad general interest.
>
> 3. This was the result of a misplaced latex \label tag, we regret this error and thank the reviewer for bringing it to our attention.

---

### Official Review · Reviewer_9YZg · 2023-10-20

**Soundness:** 3 good
**Presentation:** 3 good
**Contribution:** 2 fair
**Rating:** 3
**Confidence:** 4

**Summary:**

This paper compares semi-supervised learning (semi-SL) with self-supervised learning (self-SL) to establish when one should be used in place of the other in the context of 4 applications where unlabeled data is available: (1) In-domain, where labeled and unlabeled data sampled from same distribution, (2) near-domain, where the downstream task contains data with significant overlap as ImageNet and can use retrieval-based classification with previously-learnt features, (3) Out-of-distribution detection, and (4) natural corruption & adversarial robustness. Self-SL and semi-SL are described to differ mainly by the timeline at which unlabeled data is introduced as well as how the two frameworks select clusters. For each of self-SL and semi-SL, the authors choose 2 algorithms each and compares the performance on all of these tasks. Overall, semi-SL is shown to outperform self-SL on all tasks, or at worst performs comparably.

**Strengths:**

* With the increasing availability of pre-trained models, it is important to understand when semi-SL is beneficial. The experiments in this paper show that at least in the tasks considered, semi-SL still has an advantage over self-SL.
* The paper does a good job of describing the similarities and differences between semi-SL and self-SL.

**Weaknesses:**

* As the authors mention, self-SL is typically used to learn "generalizable representations" and is often applied when there is a very large unlabeled dataset. Further, semi-SL is typically benchmarked against in-domain performance which, despite the names of the tasks considered, is what the paper focuses mostly on. Therefore, I think it's not too surprising that semi-SL outperforms self-SL on the set of tasks the authors focus on.
* What I think would be more interesting is to understand (1) if semi-SL is still beneficial when using pre-trained models and (2) when one outperforms the other with respect to the unlabeled dataset's size. I.e. I don't think the set of tasks considered in this paper is the truly interesting ones that are relevant to practitioners.
* While this paper shows that semi-SL outperforms self-SL, it doesn't really show *why* semi-SL outperforms self-SL. The closest this paper gets to explaining the observations is some intuitions or assumptions, but I think the authors should go a bit deeper, e.g. check if we can control the timeline at which unlabeled data is introduced or if the difference in how clusters are selected is truly impacting performance.

I'm open to discussion and think the problem addressed is very important, but at this stage I feel that there's not much "new" information that can be inferred from the experimental conclusions.

**Questions:**

* Comments under weaknesses: Why not use both semi-SL and self-SL? What are possible explanations for why semi-SL outperforms self-SL in these experiments?
* What are the reasons for choosing the 2 semi-SL and 2 self-SL methods? Could the conclusions of this paper change if one chooses to use a different algorithm for semi-SL or for self-SL?
* (Minor) Certain paragraphs could be broken up into smaller paragraphs with more relevant sentences together. There are some typos (Semil-SL after Eq. (2)), and the equation rendering could be improved. In-text references should appropriately use (Author, Year) vs. Author, year using the appropriate \cite command variants.

---

> ### Author Response · Authors · 2023-11-23
>
> > I think it's not too surprising that semi-SL outperforms self-SL on the set of tasks the authors focus on
>
> We believe it is a somewhat novel finding that presence of class labels for ImageNet during training helps on tasks like Oxford and Paris retrieval which have no class overlap with ImageNet.
>
> > I don't think the set of tasks considered in this paper is the truly interesting ones that are relevant to practitioners
>
> We would respectfully disagree with this assertion, in the context of representation learning, the transferability of the representation and its robustness to different situations is of significant practical importance.

---

### Official Review · Reviewer_TpWL · 2023-10-22

**Soundness:** 2 fair
**Presentation:** 2 fair
**Contribution:** 2 fair
**Rating:** 3
**Confidence:** 5

**Summary:**

This paper attempts to analyze both self-supervised learning and semi-supervised learning at the same time. The author does a lot of experiments to try to show that semi-supervised learning is useful. But there are many parts of the expression that are puzzling, which I will expand on below.

**Strengths:**

The perspective of this paper is a thoughtful, as both self-supervised learning and semi-supervised learning are indeed paradigms that can take full advantage of unlabeled data when there is insufficient labeled data.

**Weaknesses:**

1.This paper does not reveal the relationship between self-supervised learning  and semi-supervised learning on a theoretical level.

2. All experiments should add results with only labeled data.

3.There is a problem with the citation formatting in this paper.

**Questions:**

1. As for uniformity and alignment in self-supervised learning, how is it reflected in semi-supervised learning, and what does it do for semi-supervised learning; traditional semi-supervised learning loss is rarely used clustering obj, please explain in detail in combination with fixmatch, pseudo label and other methods.

2. In self-supervised learning, there is a class called supervised contrastive  learning, whose definition of clustering is also on class-level, has this situation been considered in the article? In the case there is also a direct illustration of the early introduction of labels.

3.For experiments with OOD, closed-set acc should be reported in addition to AUROC, which are two different measurement dimensions.

4.The experiments on self-supervised learning and semi-supervised learning seem pretty cut and dry, and I don't doubt that both self-supervised learning and semi-supervised learning can be done well on tasks like in-domain, out-of-distribution data, but what does putting the results of these experiments together tell us about what's going on, and where's the connection?

**Details Of Ethics Concerns:**

None.

---

> ### Author Response · Authors · 2023-11-23
>
> > closed-set acc should be reported in addition to AUROC
>
> We report closed set accuracy for all models in Table 1.
>
> > what does putting the results of these experiments together tell us about what's going on, and where's the connection
>
> We broadly compare Semi-SL and Self-SL methods on the questions of robustness and transferability. Each of our experiments tests one aspect of robustness or transferability.

---

### Official Review · Reviewer_UA6u · 2023-11-07

**Soundness:** 3 good
**Presentation:** 4 excellent
**Contribution:** 2 fair
**Rating:** 5
**Confidence:** 4

**Summary:**

This paper explores how the choice of Self-SL versus Semi-SL framework affects performance on in-domain, near-domain and out-of-distribution data, robustness to image corruptions and adversarial attacks, cross-domain few-shot learning, and ability to learn from imbalanced data using two semi-supervised learning algorithms and two self-supervised learning algorithms.

**Strengths:**

1. The topic of the paper is attractive, and the chosen research direction is worth exploring.
2. The writing of the paper is well-done, making it easy for readers to follow and finish quickly.
3. The author has taken a comprehensive approach to experimental settings.

**Weaknesses:**

1. There are too few comparative algorithms in this paper. The author uses two algorithms each to represent semi-supervised learning and self-supervised learning, which is destined to yield less comprehensive conclusions.
2. The contribution seems somewhat limited. Apart from the analysis of experimental results, the paper appears to lack new insights. The scope of the topic is broad, but the actual work done is relatively minimal.
3. As an experimental paper without new theoretical, algorithmic, or conceptual contributions, there must be stringent requirements for the experiments. However, the current volume of experiments falls far short of the standards for an experimental paper. In addition to the issue of a limited number of comparative algorithms, there are several other concerns, such as the use of only one dataset for some settings, a narrow range of evaluation metrics, and seemingly running each experiment only once. The research direction of the paper is meaningful, but the amount of work and contributions fall far below the average standards for acceptance. Drawing significant conclusions from insufficient experiments is overly one-sided.
4. The first 5 pages of the 9-page paper are devoted to summarizing and introducing the background, while halt of the last 4 pages are occupied by figures and tables. The remaining text primarily focuses on describing experimental settings, with only a small portion dedicated to vague summarization and analysis.

**Questions:**

I feel that not all semi-supervised learning algorithms and self-supervised learning algorithms can be generalized to Equation (1). Many algorithms do not involve clustering, and in most cases, the clusters are unknown. Please provide a more detailed explanation for this.

---

> ### Author Response · Authors · 2023-11-23
>
> We thank the reviewer for noting the comprehensive nature of our experiments.
>
> We would like to note the following points about the weaknesses listed by the reviewer:
>
> 1. We choose the methods carefully to ensure comparability of results is due to the the differences in Semi- vs Self- supervised learning and not other extraneous factors. e.g. PAWS is inspired by SwAV. So its not just about 2 sets of methods vs other 2.
>
> 2. We are the first to report the significant robustness advantage enjoyed by Semi-Supervised methods over Self-SL methods. Additionally, we establish that Semi-SL methods are at least equally as transferable as SelfSL methods which is novel as well.
>
> 3. Many of the experiments we report, e.g. training free transfer to near domain datasets are deterministic and hence do not require multiple runs. Following the reviewer's suggestion, we will update results for adversarial attack and cross domain transfer to report run-to-run variance.

---

### Meta-Review · Area_Chair_6SWL · 2023-12-05

**Metareview:**

This paper compares the performance of semi-supervised learning algorithms with self-supervised learning algorithms under various data settings, including performance on in-domain, near-domain, and out-of-distribution data, robustness to image corruptions and adversarial attacks, cross-domain few-shot learning, and ability to learn from imbalanced data.  This paper proposes a valuable problem and considers many scenarios. However, there still are consistent concerns about the sufficiency of the experiments. As an experimental paper, the chosen baseline methods are not comprehensive enough to represent Semi-Supervised Learning and Self-Supervised Learning. Additionally, the current analysis is relatively simple and does not provide many new insights. In this work, the authors consider a wide range of data distribution variations to support their claim. However, these diverse data settings may not be simply summarized by the transferability and robustness of representations alone. The current manuscript gives the impression that the authors simply ran the methods in different data settings, especially in some cases where the experiments lack more methods and validation on benchmark datasets, making the arguments insufficient to support the overall claim. Therefore, I recommend rejecting this paper, but I encourage the authors to use feedback from reviewers to further improve the paper.

**Justification For Why Not Higher Score:**

The reason for not assigning a higher score is due to the following shortcomings identified in this paper:
1. The analysis of the comparison between self-supervised learning and semi-supervised learning in this paper is simplistic and lacks depth.
2. Both self-supervised and semi-supervised algorithms exhibit significant diversity, but the chosen comparative algorithms in this paper are not comprehensive enough.
3. The experimental section of this paper requires further enrichment.

**Justification For Why Not Lower Score:**

N/A

---

### Decision · Program_Chairs · 2024-01-16

Reject